# Associations between cesarean delivery and child mortality: A national record linkage longitudinal study of 17.8 million births in Brazil

Enny S. Paixao[1,2]*, Christian Bottomley[1], Julia M. Pescarini[1,2], Kerry L. M. Wong[1], Luciana L. Cardim[2], Rita de Cássia Ribeiro Silva[2,3], Elizabeth B. Brickley[1], Laura C. Rodrigues[1,2], Flavia Jôse Oliveira Alves[2], Maria do Carmo Leal[4], Maria da Conceicao N. Costa[2,5], Maria Gloria Teixeira[2,5], Maria Yury Ichihara[2], Liam Smeeth[1], Mauricio L. Barreto[2], Oona M. R. Campbell[1]

1 Infectious Disease Department, Faculty of Epidemiology and Population Health, London School of Hygiene & Tropical Medicine, London, United Kingdom, 2 Center for Data and Knowledge Integration for Health, Oswaldo Cruz Foundation, Salvador, Brazil, 3 Nutrition School, Federal University of Bahia, Salvador, Brazil, 4 Sergio Arouca National School of Public Health, Oswaldo Cruz Foundation, Rio de Janeiro, Brazil, 5 Collective Health Institute, Federal University of Bahia, Salvador, Brazil

* enny.cruz@lshtm.ac.uk

**Data Availability Statement:** All data supporting the findings presented here were obtained from Centro de Integração de Dados e Conhecimentos

## Abstract

### Background

There is an increasing use of cesarean delivery (CD) based on preference rather than on medical indication. However, the extent to which nonmedically indicated CD benefits or harms child survival remains unclear. Our hypothesis was that in groups with a low indication for CD, this procedure would be associated with higher child mortality and in groups with a clear medical indication CD would be associated with improved child survival chances.

### Methods and findings

We conducted a population-based cohort study in Brazil by linking routine data on live births between January 1, 2012 and December 31, 2018 and assessing mortality up to 5 years of age. Women with a live birth who contributed records during this period were classified into one of 10 Robson groups based on their pregnancy and delivery characteristics. We used propensity scores to match CD with vaginal deliveries (1:1) and prelabor CD with unscheduled CD (1:1) and estimated associations with child mortality using Cox regressions. A total of 17,838,115 live births were analyzed. After propensity score matching (PSM), we found that live births to women in groups with low expected frequencies of CD (Robson groups 1 to 4) had a higher death rate up to age 5 years if they were born via CD compared with vaginal deliveries (HR = 1.25, 95% CI: 1.22 to 1.28; $p < 0.001$). The relative rate was greatest in the neonatal period (HR = 1.39, 95% CI: 1.34 to 1.45; $p < 0.001$). There was no difference in mortality rate when comparing offspring born by a prelabor CD to those born by

para Saúde (CIDACS). Importantly, restrictions apply to the availability of these data. However, upon reasonable request and provided all ethical and legal requirements are met, the institutional data curation team can make the data available. Information on how to apply to access the data can be found at https://cidacs.bahia.fiocruz.br/.

**Funding:** EP is funded by the Wellcome Trust [Grant number 213589/Z/18/Z]. This research was funded by the Wellcome Trust [Grant number 202912/Z/16/Z]. The funders had no role in study design, data collection and analysis, decision to publish, or preparation of the manuscript.

**Competing interests:** NO authors have competing interests.

**Abbreviations:** CD, cesarean delivery; HDI, Human Development Index; LMIC, low- and middle-income country; PSM, propensity score matching; RECORD, Reporting of studies Conducted using Observational Routinely-collected Data; SIM, Sistema de Informação sobre Mortalidade; SINASC, Sistema de Informação sobre Nascimentos; VBAC, vaginal birth after cesarean; WHO, World Health Organization.

unscheduled CD. For the live births to women with a CD in a prior pregnancy (Robson group 5), the relative rates for child mortality were similar for those born by CD compared with vaginal deliveries (HR = 1.05, 95% CI: 1.00 to 1.10; $p$ = 0.024). In contrast, for live births to women in groups with high expected rates of CD (Robson groups 6 to 10), the child mortality rate was lower for CD than for vaginal deliveries (HR = 0.90, 95% CI: 0.89 to 0.91; $p$ < 0.001), particularly in the neonatal period (HR = 0.84, 95% CI: 0.83 to 0.85; $p$ < 0.001). Our results should be interpreted with caution in clinical practice, since relevant clinical data on CD indication were not available.

## Conclusions

In this study, we observed that in Robson groups with low expected frequencies of CD, this procedure was associated with a 25% increase in child mortality. However, in groups with high expected frequencies of CD, the findings suggest that clinically indicated CD is associated with a reduction in child mortality.

## Author summary

### Why was this study done?

- In many countries, cesarean delivery (CD) rates have been increasing.

- The growing use of this procedure has been partly driven by clinician and maternal preference rather than based on medical grounds.

- Unindicated CDs add to costs and potentially introduce harms; the effects of unindicated CD on infant and child health outcomes remain unclear.

### What did the researchers do and find?

- We analyzed over17.8 million live births in Brazil from 2012 to 2018. We classified each birth into one of 10 Robson groups. Then, we estimated the relative child mortality rates in the first 5 years of life by comparing CD versus vaginal delivery and prelabor CD versus unscheduled CD.

- Live births to women in Robson group 1 to 4 (groups with low expected rates of CD) who had a CD had a 25% increased mortality rate in the first 5 years of life compared with those born vaginally.

- Neither prelabor CD (compared with unscheduled CD) nor repeated CD (compared with vaginal) (Robson group 5) showed a statistically significant association with mortality up to the age of 5.

- Births with a noncephalic presentation, multiples (twins or triplets,) or preterm births (Robson groups 6 to 10) were at reduced risk of death if delivered by CD rather than vaginally.

**What do these findings mean?**

- Our study suggests that, in Brazil, CD is associated with an increased risk of child mortality unless there is a clear indication for the procedure.

- The study may help pregnant women and their providers make informed decisions as to whether CD is appropriate for them.

- We recommend further research in low- and middle-income settings to confirm the results. If confirmed, interventions targeting pregnant women, health workers, and health systems should be made to reduce the rates of unindicated CD.

## Introduction

In many countries, cesarean delivery (CD) rates have been increasing [1,2]. Although there is no scientific consensus on optimal CD rates at the population level [3], it is clear that in high- and middle-income countries, the growing use of this procedure (particularly among wealthy individuals) has been partly driven by institution, clinician, and maternal preference, rather than on medical grounds [4]. While the benefits of CD as a lifesaving procedure for both pregnant women and their offspring is well proven [2], unindicated CDs add to costs, and their effects on infant and child health outcomes remain unclear and may be harmful [5].

Previous studies have shown that CD reduces intestinal gut microbiome diversity among offspring and is associated with increased risks of allergy, atopy, asthma, type 1 diabetes, and obesity [5–7]. Studies in high-income countries have explored the effects of CD on neonatal and infant mortality among women with different obstetric histories [7,8]. Data from low- and middle-income countries (LMICs), where child mortality is generally higher, are scarce [9].

Brazil has one of the world's highest CD rates (56%), and, in the private sector, the CD rate is almost 90% [10]. Brazilian vital registration data provide a unique opportunity to estimate the effect of CD on childhood mortality. However, studies of the mortality consequences of medical procedures risk confounding by the indication for the procedure. For example, babies experiencing fetal distress and delivered by cesarean section may die despite the CD [11]. At the same time, data on the indications for CD may be unreliable, as clinicians seek to justify the CD [12]. To this end, we analyzed the effect of CD stratified by the Robson classification system, also known as the Ten Group Classification System.

The Robson classification system (endorsed by the World Health Organization (WHO), the International Federation of Gynecology and Obstetrics, and the European Board of Obstetrics and Gynaecology) [13] groups women into one of 10 mutually exclusive categories, based on 6 obstetric characteristics: parity, previous CD, gestational age, type of onset of labor, fetal presentation, and number of fetuses. According to WHO, we should expect a low level of clinical need, and a low CD rate, in groups 1 to 4 [14] (women with a term, cephalic presentation, and singleton fetus). In contrast, we should expect a higher level of need and a higher CD rate in group 5 (women with a previous CD) and in groups 6 to 10 (women with twins, breech, other abnormal presentation, or preterm birth).

Using data from more than 17.8 million births, we investigated the association between CD and child mortality according to the Robson classification and explored the relationship between pre- and postlabor CD and death rate in offspring. We hypothesized that in groups with a low indication for CD, this procedure would be associated with higher child mortality.

On the other hand, in groups with a clear medical indication, we hypothesized that CD would be associated with improved child survival chances.

## Methods

### Study design

We conducted a population-based cohort study by linking routine data on live births from January 1, 2012 to December 31, 2018 in Brazil with records on death. These live births were followed up from birth until December 31, 2018, death, or up to the age of 5 years.

### Data source

Data were extracted from the Brazil Live Birth Information System (Sistema de Informação sobre Nascimentos, SINASC) and the Mortality Information System (Sistema de Informação sobre Mortalidade, SIM). Live birth records are legally required and are completed by the health worker who assisted the childbirth. SINASC records include the mother's name, place of residence, age, marital status, education, maternal race/ethnicity obstetric history (previous CD or vaginal deliveries), and pregnancy characteristics. The latter include length of gestation, type of delivery, fetal presentation, delivery onset (prelabor CD, induced, or spontaneous vaginal delivery), and characteristics of the neonate (twins and other multiples, birth weight, and presence of congenital anomalies) [15]. The SINASC form does not record the number of previous births, so we used the number of previous pregnancies as a proxy for parity. All data items were over 85% complete, except for previous CD (82%). An evaluation of birth registration data found that over 97% of Brazilian live births were registered [16].

Death certificates are also legally required and are completed by a physician. SIM records include information on the deceased (name, place of residence, age, marital status, education, date, and cause of death) and, for children, information on the deceased's mother. Among infant death records, all items were over 85% complete, except for information on length of pregnancy (81%) and maternal occupation (78%) [17].

### Linkage process

We linked SINASC live births records with deaths registered in SIM. The matching variables were the name of mother, maternal age at birth, maternal date of birth, and the municipality of residence of the mother at the time of delivery. We excluded duplicate records and those with missing or implausible names. The linkage was performed with CIDACS-RL-Record Linkage, a novel record linkage tool developed to link large-scale administrative datasets at the CIDACS [18]. According to ethical and legal rules, linkage procedures were conducted at CIDACS in a strict data protection environment [19].

### Procedures

Once data were linked, all records were eligible to be part of the study. We excluded records with contradictory data (e.g., simultaneously reporting no previous pregnancy and a previous vaginal delivery) and records with incomplete information on the mode of delivery, previous pregnancy, gestational weeks at delivery, number of fetuses, delivery onset, and a previous CD. We then classified each record into one of the 10 Robson groups. Robson groups were then grouped into those with a low expected CD rate (groups 1 to 4) and those with a high expected rate of CD (groups 6 to 10). Robson group 5, births to women with a previous history of CD, was kept separate.

## Statistical analyses

We used Kaplan–Meier estimates of mortality risk and Cox regression to compare mortality in the first 5 years of life by cesarean versus vaginal mode of delivery using the age of the child in days since birth for the timescale in our survival analyses. Statistical significance was defined as 95% CIs that excluded 1.0, with $p$-values calculated accordingly The analyses were done separately for each Robson group, for the combined groups 1 to 4 (excluding 2b and 4b because they were all prelabor CD) (low expected CD rates), and 6 to 10 (high expected CD rates), and for the cohort as a whole (excluding 2b and 4b). We also compared prelabor CD versus unscheduled CD. In this analysis, groups 2b and 4b (nullipara and multipara with prelabor CD) were compared with those who had a CD from groups 2a and 4a (attempted labor, with unscheduled CD). In each analysis, we used propensity score matching (PSM) to control for confounding. The propensity score was obtained via logistic regression. Matching—on the logit of the propensity score—was done using a nearest neighbor algorithm matching (1:1) without replacement, and with a caliper of 0.1, as recommended by Austin [20]. Because vaginal birth was more common than CD in Robson groups 1 to 4, we generated matched pairs by selecting a vaginal birth for each CD. Conversely, for groups 5 to 10, where CD was more common, we selected a CD for each vaginal birth. In the analysis of the entire cohort, we selected a vaginal delivery for each CD. The propensity score model matched on potential confounders available in the SINASC dataset: maternal age (in 2-year groups), level of maternal education (none, 1 to 3, 4 to 7, 8 to 12, and more than 12 years of schooling), self-declared maternal race/ethnicity (White, Black, Asian, Mixed race, and Indigenous), marital status (single, widowed, divorced, and married/union), Human Development Index (HDI) of the municipality of residence of the mother (s≥0.80, <0.80 to ≥0.70, <0.70 to ≥55, and <0.55), number of prenatal care appointments (none, 1 to 3, 4 to 6, and more than 7 appointments), sex of newborn (female or male), birth weight (in 200-g categories), year of birth, and Robson group (1 to 10). Interaction terms were not included in the models. To assess the robustness of the PSM, we also conducted an alternative analysis using a conventional Cox proportional hazards model adjusted for the same confounders used to estimate the propensity score.

We undertook additional analyses to test our data. First, to further assess whether there might have been a CD indication in groups 1 to 4 that subsequently increased the death rate, we conducted analyses of child mortality, conditional on survival up to 6, 27, and 364 days. Second, we investigated the potential for residual confounding by applying the propensity score analysis to child deaths from external causes (e.g., transport accidents, homicides, and accidental injuries). Since deaths from external causes are more common in low socioeconomic status groups, but are unlikely to be affected by CD, they can be used to assess how well matching reduces socioeconomic differences between the exposure groups.

We did not have a written analysis plan; however, we specified the list of confounders and categorizations of the continuous variables before analyzing the data. All analyses were done using STATA version 15.0. This study is reported as per the Reporting of studies Conducted using Observational Routinely-collected Data (RECORD) guideline (Table A in S1 Text).

## Ethics statement

Ethical approval was obtained from the Federal University of Bahia's Institute of Public Health Ethics Committee (CAAE registration number: 18022319.4.0000.5030) and the London School of Hygiene & Tropical Medicine reference number 22817.

## Results

During the study period, 20,526,629 live births were registered in the SINASC. Of those, 17,838,115 (86.9%) had sufficient information to be classified into one of the Robson groups

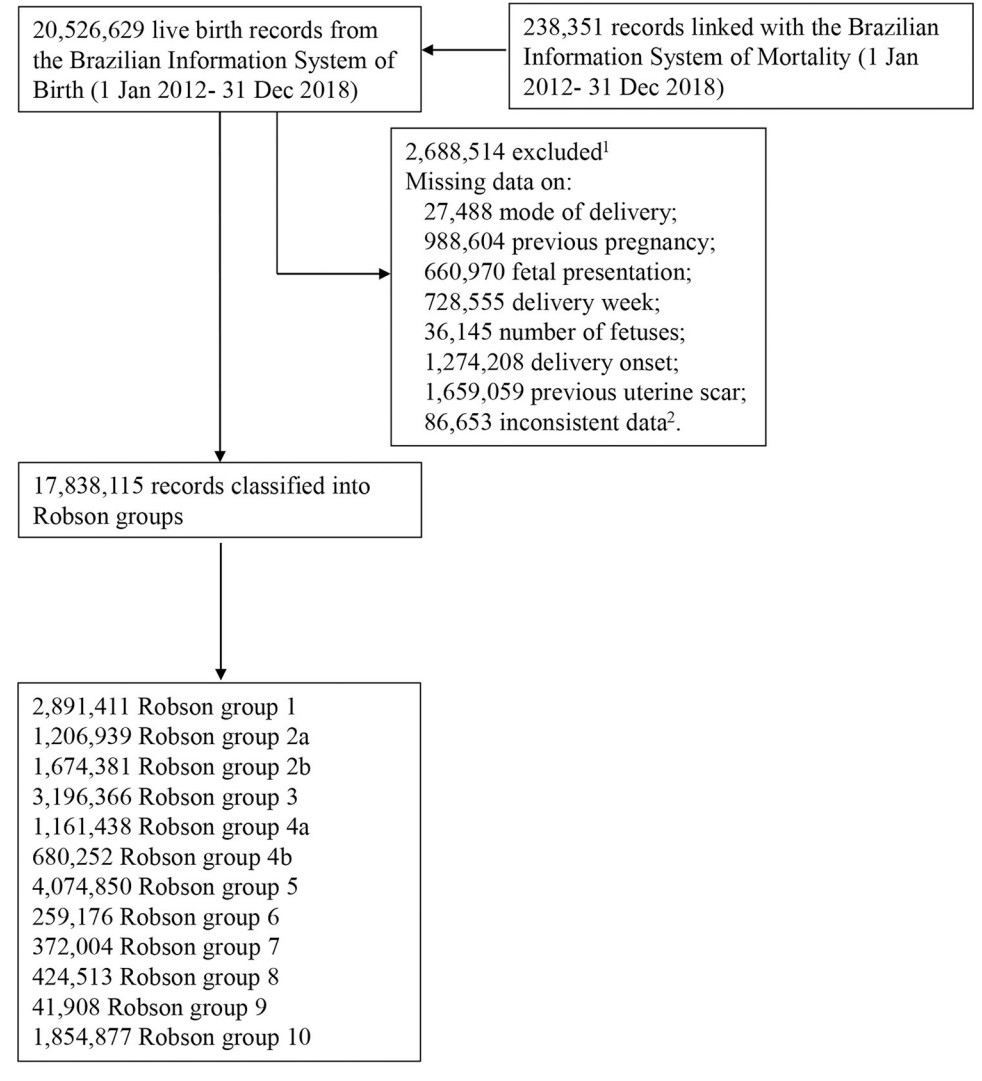

[1]The reasons for exclusion do not sum to 4,225,844 because a record may be missing multiple variables.

[2]Contradictory data, such as records with no previous pregnancy and a previous vaginal delivery

**Fig 1. Flowchart of cohort selection process and Robson classification.**

(Fig 1). The proportion of live births via CD varied by Robson group, from 11.3% in group 4a to 100% in groups 2b and 4b (Table A in S2 Text). The characteristics of live births delivered vaginally or by CD are described in Table 1. In Table 2, we compare live births stratified by Robson group and mode of delivery. In general, live births via CD had mothers who were older, more educated, more likely to be White, and who had more antenatal care appointments and were delivered in wealthier municipalities (Table 2). However, after PSM, the 2 groups had very similar baseline characteristics; the standardized mean difference between both

**Table 1.** Characteristics (number and percent) of 17,838,115 live births deliveries by mode of delivery in Brazil from 2012 to 2018.

| | Vaginal delivery | | CD | |
|---|---|---|---|---|
| | *n* | % | *n* | % |
| **Robson category** | | | | |
| Robson 1 | 1,553,898 | 53.77 | 1,336,141 | 46.23 |
| Robson 2a | 896,049 | 74.28 | 310,326 | 25.72 |
| Robson 2b | 0 | 0.00 | 1,674,370 | 100 |
| Robson 3 | 2,576,452 | 80.65 | 618,340 | 19.35 |
| Robson 4a | 1,028,958 | 88.65 | 131,800 | 11.35 |
| Robson 4b | 0 | 0.00 | 680,235 | 100.00 |
| Robson 5 | 608,012 | 14.03 | 3,724,268 | 85.97 |
| Robson 6 | 24,911 | 9.62 | 234,081 | 90.38 |
| Robson 7 | 53,920 | 14.51 | 317,616 | 85.49 |
| Robson 8 | 71,552 | 16.87 | 352,631 | 83.13 |
| Robson 9 | 1,318 | 3.15 | 40,564 | 96.85 |
| Robson 10 | 920,625 | 49.66 | 933,116 | 50.34 |
| **Maternal age** | | | | |
| <20 years | 1,832,970 | 59.99 | 1,222,596 | 40.01 |
| 20 to 24 years | 2,235,808 | 50.36 | 2,203,997 | 49.64 |
| 25 to 29 years | 1,739,093 | 40.28 | 2,578,524 | 59.72 |
| 30 to 34 years | 1,188,146 | 32.92 | 2,420,598 | 67.08 |
| 35 to 39 years | 569,440 | 29.60 | 1,354,430 | 70.40 |
| 40 to 44 years | 136,040 | 29.89 | 319,110 | 70.11 |
| 45+ years | 9,148 | 31.22 | 20,152 | 68.78 |
| **Marital status** | | | | |
| Single | 3,759,845 | 50.77 | 3,645,244 | 49.23 |
| Widow | 3,795,177 | 37.86 | 6,229,019 | 62.14 |
| Divorced | 12,967 | 39.98 | 19,468 | 60.02 |
| Married/union | 65,389 | 31.37 | 143,056 | 68.63 |
| **Maternal education** | | | | |
| None | 65,574 | 71.06 | 26,706 | 28.94 |
| 1 to 3 years | 301,334 | 61.88 | 185,656 | 38.12 |
| 4 to 7 years | 1,870,591 | 57.37 | 1,389,820 | 42.63 |
| 8 to 12 years | 4,726,488 | 45.08 | 5,757,360 | 54.92 |
| 12 + years | 646,277 | 19.64 | 2,644,979 | 80.36 |
| **Maternal ethnicity** | | | | |
| White | 2,206,641 | 32.93 | 4,495,154 | 67.07 |
| Black | 491,661 | 50.45 | 482,877 | 49.55 |
| Asian | 29,985 | 41.86 | 41,646 | 58.14 |
| Mixed race | 4,647,470 | 49.36 | 4,768,158 | 50.64 |
| Indigenous | 109,749 | 79.48 | 28,337 | 20.52 |
| **Year of birth** | | | | |
| 2012 | 1,008,611 | 43.42 | 1,314,091 | 56.58 |
| 2013 | 1,032,185 | 42.35 | 1,405,269 | 57.65 |
| 2014 | 1,084,318 | 42.21 | 1,484,397 | 57.79 |
| 2015 | 1,144,394 | 43.73 | 1,472,368 | 56.27 |
| 2016 | 1,115,905 | 43.80 | 1,431,908 | 56.20 |
| 2017 | 1,147,956 | 43.71 | 1,478,324 | 56.29 |
| 2018 | 1,177,415 | 43.44 | 1,533,050 | 56.56 |

*(Continued)*

**Table 1.** (Continued)

| | Vaginal delivery | | CD | |
|---|---|---|---|---|
| | *n* | % | *n* | % |
| **Sex of the newborn** | | | | |
| Female | 3,812,970 | 43.85 | 4,882,133 | 56.15 |
| Male | 3,896,651 | 42.67 | 5,235,424 | 57.33 |
| **Number of prenatal appointments** | | | | |
| None | 78,315 | 74.61 | 26,654 | 25.39 |
| 0 to 3 appointments | 728,435 | 64.74 | 396,803 | 35.26 |
| 4 to 6 appointments | 2,249,768 | 52.38 | 2,045,519 | 47.62 |
| 7+ appointments | 4,509,614 | 37.48 | 7,521,203 | 62.52 |
| **Birth weight (grams)** | | | | |
| <1,500 | 103,742 | 42.35 | 141,204 | 57.65 |
| 1,500 to 1,999 | 104,867 | 34.88 | 195,826 | 65.12 |
| 2,000 to 2,499 | 440,191 | 43.07 | 581,872 | 56.93 |
| 2,500 to 2,999 | 1,908,038 | 46.76 | 2,172,773 | 53.24 |
| 3,000 to 3,499 | 3,264,252 | 44.32 | 4,101,054 | 55.68 |
| 3,500 to 3,999 | 1,581,428 | 40.56 | 2,318,005 | 59.44 |
| 4,000+ | 298,763 | 33.07 | 604,604 | 66.93 |
| **Municipality HDI** | | | | |
| Very high | 1,570,448 | 40.80 | 2,278,504 | 59.20 |
| High | 3,942,065 | 39.60 | 6,012,422 | 60.40 |
| Medium | 1,990,053 | 53.08 | 1,759,331 | 46.92 |
| Low | 208,188 | 75.07 | 69,150 | 24.93 |

CD, cesarean delivery; HDI, Human Development Index.

groups was less than 0.07 on all covariates (Tables B–D in S2 Text). The coefficients of the propensity score models are shown in Table E in S2 Text, and the distribution of scores is shown in S1 Fig.

The crude child mortality risk varied significantly by Robson groups as well as by the mode of delivery. Robson groups 6 to 10 had the highest mortality risks. Robson group 6 had the highest mortality risk among those delivered vaginally and the largest difference with those delivered via cesarean, 131.5 per 1,000 live births versus 2.0 per 1,000 live births. This was followed by group 8, which was 91.1 per 1,000 live births versus 24.1 per 1,000 live births, respectively. The mortality risks between vaginal and cesarean deliveries were not statistically significant different in groups 1 to 4 (Table 3).

## Robson groups 1 to 4 (live births to (nulliparous or multiparous) women without a previous CD, at term, with a singleton, and cephalic baby)

In Robson groups 1 to 4 (excluding 2b and 4b) (Table 3, Fig 2), there was no difference in mortality rate up to the age of 5 years between those born via CD compared with those born vaginal [1.74 versus 1.74 deaths/1,000 person-years (HR = 1.00, 95% CI: 0.99 to 1.03; $p = 0.487$) before adjusting via PSM. After adjusting via PSM, children born via CD were significantly more likely to die in the first 5 years of life (HR = 1.25, 95% CI: 1.22 to 1.28; $p < 0.001$); 39% more likely to die within the first 28 days (HR = 1.39, 95% CI: 1.34 to 1.45; $p < 0.001$); and 29% more likely to die within the first year (HR = 1.29, 95% CI: 1.25 to 1.33; $p < 0.001$).

**Table 2. Characteristics (number and percent) of 17,838,115 live births deliveries in Brazil from 2012 to 2018 by mode of delivery.**

| | Robson 1 to 4 | | Robson 5 | | Robson 6 to 10 | |
|---|---|---|---|---|---|---|
| | Vaginal delivery | CD | Vaginal delivery | CD | Vaginal delivery | CD |
| **Robson category** | | | | | | |
| Robson 1 | 1,553,898 (53.77) | 1,336,141 (46.23) | - | - | - | - |
| Robson 2a | 896,049 (74.28) | 310,326 (25.72) | - | - | - | - |
| Robson 2b | 0 (0.0) | 1,674,370 (100) | | | | |
| Robson 3 | 2,576,452 (80.65) | 618,340 (19.35) | - | - | - | - |
| Robson 4a | 1,028,958 (88.65) | 131,800 (11.35) | - | - | - | - |
| Robson 4b | 0 (0.0) | 680,235 (100) | | | | |
| Robson 5 | - | - | 583,101 (14.32) | 3,490,187 (85.68) | - | - |
| Robson 6 | - | - | - | - | 24,911 (9.62) | 234,081 (90.38) |
| Robson 7 | - | - | - | - | 53,920 (14.51) | 317,616 (85.49) |
| Robson 8 | - | - | - | - | 71,552 (16.87) | 352,631 (83.13) |
| Robson 9 | - | - | - | - | 1,318 (3.15) | 40,564 (96.85) |
| Robson 10 | - | - | - | - | 920,625 (49.66) | 933,116 (50.34) |
| **Maternal age** | | | | | | |
| <20 years | 1,485,483 (64.17) | 829,310 (35.83) | 45,241 (22.92) | 152,137 (77.08) | 302,246 (55.62) | 241,149 (44.38) |
| 20 to 24 years | 1,791,228 (60.40) | 1,174,465 (39.60) | 159,230 (19.70) | 649,012 (80.30) | 285,350 (42.85) | 380,520 (57.15) |
| 25 to 29 years | 1,358,894 (53.15) | 1,198,048 (46.85) | 166,724 (15.12) | 936,208 (84.88) | 213,475 (32.46) | 444,268 (67.54) |
| 30 to 34 years | 900,664 (47.76) | 985,036 (52.24) | 128,183 (11.53) | 983,750 (88.47) | 159,299 (26.07) | 451,812 (73.93) |
| 35 to 39 years | 415,400 (47.66) | 456,122 (52.34) | 67,232 (9.81) | 618,163 (90.19) | 86,808 (23.66) | 280,145 (76.34) |
| 40 to 44 years | 97,006 (48.90) | 101,373 (51.10) | 15,655 (9.81) | 143,972 (90.19) | 23,379 (24.07) | 73,765 (75.93) |
| 45+ years | 6,584 (48.98) | 6,858 (51.02) | 836 (10.74) | 6,945 (89.26) | 1,728 (21.39) | 6,349 (78.61) |
| **Marital status** | | | | | | |
| Single | 2,966,687 (61.78) | 1,835,048 (38.22) | 264,352 (19.25) | 1,108,773 (80.75) | 528,806 (42.98) | 701,423 (57.02) |
| Widow | 2,971,778 (51.27) | 2,824,219 (48.73) | 303,813 (11.78) | 2,275,825 (88.22) | 519,586 (31.52) | 1,128,975 (68.48) |
| Divorced | 9,805 (58.09) | 7,074 (41.91) | 1,295 (13.18) | 8,527 (86.82) | 1,867 (32.56) | 3,867 (67.44) |
| Married/union | 48,743 (49.95) | 48,848 (50.05) | 8,258 (10.98) | 66,926 (89.02) | 8,388 (23.52) | 27,282 (76.48) |
| **Maternal education** | | | | | | |
| None | 48,965 (81.61) | 11,034 (18.39) | 4,241 (33.96) | 8,246 (66.04) | 12,368 (62.48) | 7,426 (37.52) |
| 1 to 3 years | 224,061 (76.36) | 69,378 (23.64) | 25,122 (26.07) | 71,247 (73.93) | 52,151 (53.66) | 45,031 (46.34) |
| 4 to 7 years | 1,427,180 (72.24) | 548,515 (27.76) | 153,981 (21.96) | 547,339 (78.04) | 289,430 (49.61) | 293,966 (50.39) |
| 8 to 12 years | 3,772,446 (57.96) | 2,736,785 (42.04)) | 342,120 (14.73) | 1,979,900 (85.27) | 611,922 (37.03) | 1,040,675 (62.97) |
| 12 + years | 507,375 (27.52) | 1,336,041 (72.48) | 50,851 (5.69) | 843,321 (94.31) | 88,051 (15.90) | 465,617 (84.10) |
| **Maternal ethnicity** | | | | | | |
| White | 1,739,617 (44.52) | 2,168,266 (55.48) | 184,482 (10.74) | 1,532,594 (89.26) | 282,542 (26.24) | 794,294 (73.76) |
| Black | 386,903 (63.87) | 218,902 (36.13) | 38,371 (18.74) | 166,425 (81.26) | 66,387 (40.50) | 97,550 (59.50) |
| Asian | 23,993 (54.69) | 19,877 (45.31) | 2,045 (12.80) | 13,933 (87.20) | 3,947 (33.50) | 7,836 (66.50) |
| Mixed race | 3,651,191 (62.29) | 2,210,210 (37.71) | 336,532 (16.90) | 1,654,824 (83.10) | 659,747 (42.21) | 903,124 (57.79) |
| Indigenous | 84,390 (86.57) | 13,093 (13.43) | 5,283 (38.50) | 8,438 (61.50) | 20,076 (74.68) | 6,806 (25.32) |
| **Year of birth** | | | | | | |
| 2012 | 779,422 (55.09) | 635,508 (44.91) | 66,047 (13.74) | 414,718 (86.26) | 163,142 (38.21) | 263,865 (61.79) |
| 2013 | 805,525 (54.14) | 682,387 (45.86) | 69,905 (13.27) | 456,735 (86.73) | 156,755 (37.07) | 266,147 (62.93) |
| 2014 | 853,197 (54.22) | 720,330 (45.78) | 76,177 (13.40) | 492,299 (86.60) | 154,944 (36.31) | 271,768 (63.69) |
| 2015 | 901,913 (56.62) | 691,144 (43.38) | 87,637 (14.59) | 512,932 (85.41) | 154,844 (36.59) | 268,292 (63.41) |
| 2016 | 876,859 (56.95) | 662,822 (43.05) | 89,068 (15.01) | 504,317 (84.99) | 149,978 (36.16) | 264,769 (63.84)) |
| 2017 | 906,550 (57.47) | 670,918 (42.53) | 94,879 (14.98) | 538,701 (85.02) | 146,527 (35.29) | 268,705 (64.71) |
| 2018 | 931,891 (57.52) | 688,103 (42.48) | 99,388 (14.84) | 570,485 (85.16) | 146,136 (34.74) | 274,462 (65.26) |

(*Continued*)

**Table 2.** (Continued)

| | Robson 1 to 4 | | Robson 5 | | Robson 6 to 10 | |
|---|---|---|---|---|---|---|
| | **Vaginal delivery** | **CD** | **Vaginal delivery** | **CD** | **Vaginal delivery** | **CD** |
| **Sex of the newborn** | | | | | | |
| Female | 3,011,634 (57.06) | 2,266,815 (42.94) | 293,561 (14.73) | 1,699,421 (85.27) | 507,775 (35.67) | 915,897 (64.33) |
| Male | 3,043,206 (55.06) | 2,483,924 (44.94) | 289,490 (13.92) | 1,790,406 (86.08) | 563,955 (36.98) | 961,094 (63.02) |
| **Number of prenatal appointments** | | | | | | |
| None | 51,355 (86.63) | 7,926 (13.37) | 7,700 (43.37) | 10,055 (56.63) | 19,260 (68.95) | 8,673 (31.05) |
| 0 to 3 appointments | 476,316 (79.20) | 125,068 (20.80) | 56,641 (28.10) | 144,921 (71.90) | 195,478 (60.65) | 126,814 (39.35) |
| 4 to 6 appointments | 1,656,989 (67.21) | 808,259 (32.79) | 162,869 (18.48) | 718,398 (81.52) | 429,910 (41.31) | 518,862 (54.69) |
| 7+ appointments | 3,774,213 (50.07) | 3,763,037 (49.93) | 342,882 (11.76) | 2,573,226 (88.24) | 392,519 (24.88) | 1,184,940 (75.12) |
| **Birth weight (grams)** | | | | | | |
| <1,500 | 4,639 (53.35) | 4,057 (46.65) | 549 (16.68) | 2,743 (83.32) | 98,554 (42.31) | 134,404 (57.69) |
| 1,500 to 1,999 | 14,117 (48.41) | 15,046 (51.59) | 1,575 (18.49) | 6,941 (81.51) | 89,175 (33.91) | 173,839 (66.09) |
| 2,000 to 2,499 | 205,736 (58.29) | 147,198 (41.71) | 19,329 (20.22) | 76,252 (79.78) | 215,126 (37.51) | 358,422 (62.49) |
| 2,500 to 2,999 | 1,458,557 (58.93) | 1,016,627 (41.07) | 129,773 (16.86) | 640,134 (83.14) | 319,708 (38.26) | 516,012 (61.74) |
| 3,000 to 3,499 | 2,758,673 (56.77) | 2,100,309 (43.23) | 263,446 (14.52) | 1,550,456 (85.48) | 242,133 (34.97) | 450,289 (65.03) |
| 3,500 to 3,999 | 1,352,561 (53.60) | 1,170,772 (46.40) | 140,122 (12.83) | 951,686 (87.17) | 88,745 (31.22) | 195,547 (68.78) |
| 4,000+ | 255,443 (46.36) | 295,611 (53.64) | 27,929 (9.67) | 260,796 (90.33) | 15,391 (24.20) | 48,197 (75.80) |
| **Municipality HDI** | | | | | | |
| Very high (>0.80) | 1,246,989 (53.52) | 1,082,943 (46.48) | 136,928 (15.32) | 756,831 (84.68) | 186,531 (29.83) | 438,730 (70.17) |
| High (<0.80 to 0.70) | 3,066,036 (52.61) | 2,762,054 (47.39) | 316,653 (13.06) | 2,107,070 (86.94) | 559,376 (32.85) | 1,143,298 (67.15) |
| Medium (70 to 55) | 1,578,589 (64.42) | 872,003 (35.58) | 118,782 (16.41) | 604,848 (83.59) | 292,682 (50.89) | 282,480 (49.11) |
| Low (<55) | 163,723 (82.72) | 34,212 (17.28) | 10,735 (33.37) | 21,438 (66.63) | 33,730 (71.42) | 13,500 (28.58) |

CD, cesarean delivery; HDI, Human Development Index.

Before adjusting via PSM, singleton term babies born by unscheduled CD to nulliparous and multiparous women (2b and 4b) were more likely to die in the first 5 years of life compared with those born by planned CD (2a and 4a). In group 2 (nulliparas), the rates were 1.46 versus 1.18 deaths/1,000 person-years (HR = 1.22, 95% CI: 1.15 to 1.29; $p < 0.001$). In group 4 (multiparas), the rates were 1.99 versus 1.83 deaths/1,000 person-years (HR = 1.08, 95% CI: 1.00 to 1.17; $p = 0.026$). After adjusting via PSM, there was no significant difference in the mortality rate ratio when comparing offspring born by an unscheduled CD with those born by labor CD in group 2. In group 4, the CI were close to the null value (Table 4). In a breakdown by time period, unscheduled CD was associated with an increased mortality rate during the neonatal period (HR = 1.18, 95% CI: 1.05 to 1.32; $p = 0.003$) in group 2 and in the first year of life (HR = 1.15, 95% CI: 1.03 to 1.29; $p = 0.010$) in group 4 (Table 3).

In the conditional survival analyses, we sought to understand if there was an increased mortality rate beyond an immediate risk that might be associated with the indication for the CD. After adjusting via PSM, for combined Robson groups 1 to 4 (excluding 2b and 4b), we found an increased death rate up to age 5 years following a CD compared with vaginal birth (HR = 1.18, 95% CI: 1.14 to 1.22; $p < 0.001$) for those who survived the first 7 days after birth. For those who survived the first 28 days after birth, the death rate up to age 5 years was also higher following CD (HR = 1.14, 95% CI: 1.10 to 1.19; $p < 0.001$). Beyond 1 year of life, the increased rate associated with CD remained for group 1 (HR = 1.18, 95% CI: 1.07 to 1.29; $p < 0.001$) and combined Robson groups 1 to 4 (HR = 1.10, 95% CI: 1.04 to 1.17; $p = 0.002$) (Table F in S2 Text).

Table 3. Under-five, infant, and neonatal mortality by mode of delivery, Brazil (2012 to 2018).

| Robson groups | Mortality risk from birth to 5 years/1,000 live births* | | Mortality from birth to 5 years of age (under-five mortality) | | | | Mortality from birth to 1 year of age (infant mortality) | | | | Mortality from birth to 27 days of age (neonatal mortality) | | | |
|---|---|---|---|---|---|---|---|---|---|---|---|---|---|---|
| | | | Deaths/1,000 person-years | | PSM | | Deaths/1,000 person-years | | PSM+ | | Deaths/1,000 person-years | | PSM | |
| | Vaginal | CD | Vaginal | CD | Before HR$ (95% CI)—p-value | After HR$ (95% CI)—p-value | Vaginal | CD | Before HR$ (95% CI)—p-value | After HR$ (95% CI)—p-value | Vaginal | CD | Before HR$ (95% CI)—p-value | After HR$ (95% CI)—p-value |
| 1 | 4.77 | 4.60 | 1.71 | 1.56 | 0.94 (0.91 to 0.98) 0.001 | 1.22 (1.18 to 1.28) <0.001 | 3.45 | 4.27 | 0.96 (0.92 to 1.00) 0.051 | 1.24 (1.18 to 1.30) <0.001 | 30.01 | 32.17 | 1.08 (1.02 to 1.13) 0.003 | 1.32 (1.25 to 1.40) <0.001 |
| 2a | 4.28 | 4.50 | 1.35 | 1.46 | 1.06 (1.00 to 1.13) 0.045 | 1.21 (1.12 to 1.31) <0.001 | 3.60 | 4.09 | 1.13 (1.05 to 1.21) <0.001 | 1.22 (1.12 to 1.34) <0.001 | 24.90 | 31.04 | 1.24 (1.13 to 1.36) <0.001 | 1.25 (1.12 to 1.40) <0.001 |
| 3 | 5.66 | 6.57 | 2.01 | 2.21 | 1.13 (1.09 to 1.17) <0.001 | 1.35 (1.29 to 1.42) <0.001 | 5.23 | 6.29 | 1.20 (1.16 to 1.25) <0.001 | 1.38 (1.31 to 1.46) <0.001 | 30.69 | 46.17 | 1.50 (1.42 to 1.58) <0.001 | 1.39 (1.31 to 1.46) <0.001 |
| 4a | 4.90 | 6.13 | 1.53 | 1.99 | 1.26 (1.17 to 1.36) <0.001 | 1.35 (1.21 to 1.50) <0.001 | 4.19 | 5.68 | 1.34 (1.24 to 1.46) <0.001 | 1.34 (1.19 to 1.50) <0.001 | 25.44 | 40.92 | 1.60 (1.43 to 1.79) <0.001 | 1.51 (1.29 to 1.77) <0.001 |
| 1 to 4 (2a and 4a) | 5.18 | 5.10 | 1.74 | 1.74 | 1.00 (0.99 to 1.03) 0.487 | 1.25 (1.22 to 1.28) <0.001 | 4.60 | 4.84 | 1.05 (1.02 to 1.07) <0.001 | 1.29 (1.25 to 1.33) <0.001 | 28.76 | 36.24 | 1.26 (1.22 to 1.29) <0.001 | 1.39 (1.34 to 1.45) <0.001 |
| 5 | 5.63 | 4.59 | 2.02 | 1.59 | 0.80 (0.77 to 0.83) <0.001 | 1.05 (1.00 to 1.10) 0.024 | 5.25 | 4.35 | 0.83 (0.79 to 0.86) <0.001 | 1.09 (1.03 to 1.15) <0.001 | 33.03 | 28.88 | 0.87 (0.82 to 0.93) <0.001 | 1.12 (1.04 to 1.21) 0.002 |
| 6 | 131.50 | 1.98 | 50.10 | 6.71 | 0.13 (0.12 to 0.14) <0.001 | 0.69 (0.65 to 0.72) <0.001 | 175.64 | 21.53 | 0.12 (0.12 to 0.13) <0.001 | 0.67 (0.63 to 0.71) <0.001 | 2,013.08 | 204.64 | 0.10 (0.10 to 0.11) <0.001 | 0.61 (0.58 to 0.65) <0.001 |
| 7 | 82.54 | 24.27 | 30.85 | 8.48 | 0.27 (0.26 to 0.28) <0.001 | 0.69 (0.65 to 0.72) <0.001 | 106.36 | 26.91 | 0.25 (0.25 to 0.27) <0.001 | 0.68 (0.64 to 0.71) <0.001 | 1,168.45 | 248.62 | 0.22 (0.20 to 0.23) <0.001 | 0.61 (0.59 to 0.65) <0.001 |
| 8 | 91.09 | 24.09 | 34.92 | 8.48 | 0.24 (0.23 to 0.25) <0.001 | 0.79 (0.76 to 0.82) <0.001 | 118.91 | 27.29 | 0.23 (0.22 to 0.24) <0.001 | 0.79 (0.76 to 0.82) <0.001 | 1,279.78 | 252.74 | 0.20 (0.19 to 0.21) <0.001 | 0.73 (0.70 to 0.77) <0.001 |
| 9 | 62.21 | 26.99 | 20.77 | 8.72 | 0.40 (0.26 to 0.46) <0.001 | 0.88 (0.67 to 1.14) 0.348 | 78.20 | 29.67 | 0.38 (0.30 to 0.48) <0.001 | 0.84 (0.64 to 1.10) 0.210 | 975.64 | 303.15 | 0.31 (0.25 to 0.40) <0.001 | 0.76 (0.58 to 1.00) 0.590 |
| 10 | 46.47 | 38.29 | 16.30 | 13.19 | 0.80 (0.79 to 0.81) <0.001 | 0.86 (0.85 to 0.88) <0.001 | 54.26 | 43.82 | 0.80 (0.79 to 0.81) <0.001 | 0.86 (0.84 to 0.87) <0.001 | 560.70 | 421.67 | 0.75 (0.74 to 0.77) <0.001 | 0.81 (0.80 to 0.83) <0.001 |
| 6 to 10 | 53.26 | 30.72 | 18.88 | 10.52 | 0.55 (0.54 to 0.56) <0.001 | 0.90 (0.89 to 0.91) <0.001 | 63.82 | 34.70 | 0.54 (0.53 to 0.55) <0.001 | 0.89 (0.88 to 0.90) <0.001 | 621.74 | 305.75 | 0.50 (0.49 to 0.50) <0.001 | 0.84 (0.83 to 0.85) <0.001 |
| Overall | 11.84 | 11.09 | 4.08 | 3.81 | (0.92 to 0.94) <0.001 | 1.02 (1.01 to 1.04) <0.001 | 12.40 | 11.68 | 0.94 (0.93 to 0.95) <0.001 | 1.01 (1.00 to 1.01) 0.003 | 112.45 | 102.42 | 0.91 (0.90 to 0.92) <0.001 | 0.96 (0.94 to 0.97) <0.001 |

$ HR from a Cox regression in which vaginal deliveries are the comparison group.

* The mortality risk is lower than expected because we used the complete sample; therefore, some individuals were censored before the age of 5 years.

+ Variables used in PSM: maternal age, marital status, race/ethnicity, education, year of birth, birth weight, and sex of the newborn, number of prenatal appointments, and the HDI of the maternal municipality of residence and Robson group in combined analyses.

CD, cesarean delivery; HDI, Human Development Index; PSM, propensity score matching.

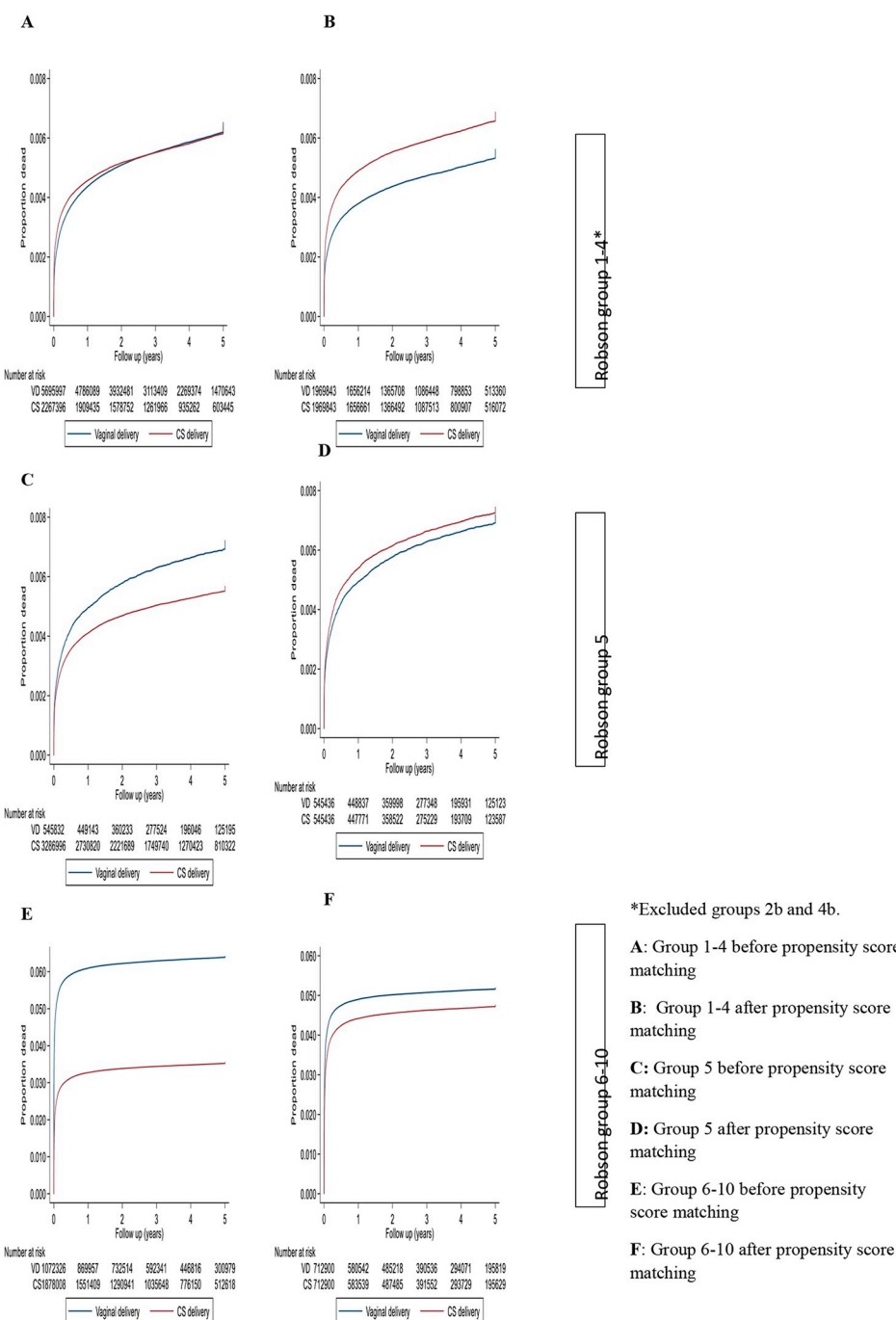

**Fig 2. Kaplan–Meier curves of mortality from birth up to 5 years for vaginal and cesarean births by Robson group.** CD, cesarean delivery.

## Robson group 5 (live births to women with a previous CD)

Before adjusting via PSM, in Robson group 5, live births born via CD were less likely to die than those born vaginally (1.59 versus 2.02 deaths/1,000 person-years, HR = 0.80, 95% CI: 0.77 to 0.83; $p < 0.001$). After adjusting via PSM, there was no significant difference in the death

**Table 4. Under-five, infant, and neonatal mortality in planned CD groups (2b and 4b) compared with unscheduled CD group 2a and 4a, Brazil (2012 to 2018).**

| Robson groups | Mortality component | Deaths/1,000 person-years | | | | PSM[+] | |
|---|---|---|---|---|---|---|---|
| | | Prelabor CD | Unscheduled CD | Before HR[$] (95% CI) | p-value | After HR[$] (95% CI) | p-value |
| 2 | **Under-five mortality** | 1.18 | 1.46 | 1.22 (1.15 to 1.29) | <0.001 | 1.04 (0.97 to 1.13) | 0.214 |
| | **Infant mortality** | 3.28 | 4.09 | 1.24 (1.16 to 1.32) | <0.001 | 1.07 (0.99 to 1.17) | 0.084 |
| | **Neonatal mortality** | 23.6 | 31.04 | 1.31 (1.20 to 1.42) | <0.001 | 1.18 (1.05 to 1.32) | 0.003 |
| 4 | **Under-five mortality** | 1.83 | 1.99 | 1.08 (1.00 to 1.17) | 0.026 | 1.13 (1.00 to 1.25) | 0.013 |
| | **Infant mortality** | 5.33 | 5.68 | 1.06 (0.98 to 1.15) | 0.130 | 1.15 (1.03 to 1.29) | 0.010 |
| | **Neonatal mortality** | 36.76 | 40.92 | 1.11 (0.99 to 1.24) | 0.059 | 1.11 (0.96 to 1.29) | 0.134 |

[$] HR from a Cox regression in which prelabor CD is the comparison group.

[+] Variables used in PSM: maternal age, marital status, race/ethnicity, education, year of birth, birth weight, and sex of the newborn, number of prenatal appointments, and the HDI of the maternal municipality of residence and Robson group in combined analyses.

CD, cesarean delivery; HDI, Human Development Index; PSM, propensity score matching.

rate up to age 5 years among live births to women with a repeat CD than those with vaginal birth after cesarean (VBAC), (HR = 1.05, 95% CI: 1.00 to 1.10; $p$ = 0.024). In a breakdown by age, CD within group 5 was associated with increased rate during the neonatal period (HR = 1.12, 95% CI: 1.04 to 1.21; $p$ = 0.002) and in the first year of age (HR = 1.09, 95% CI: 1.03 to 1.14; $p$ = 0.001) (Table 3).

## Robson groups 6 to 10 (multiple births, preterm births, and noncephalic births)

Before adjusting via PSM, in Robson groups 6 to 10 combined, the death rate up to 5 years was significantly lower following a CD than it was following a vaginal birth (10.52 versus 18.88 deaths/1,000 person-years, HR = 0.55, 95% CI: 0.54 to 0.56; $p$ < 0.001). The difference was less marked after PSM (HR = 0.90, 95% CI: 0.89 to 0.91; $p$ < 0.001). The protective association of CD was greatest during the neonatal period (HR = 0.84, 95% CI: 0.83 to 0.85; $p$ < 0.001) (Table 3).

### Sensitivity analyses

We analyzed deaths from external causes and did not observe a statistically significant difference in any Robson groups (Table G in S2 Text), suggesting that the PSM effectively matched and confirming our expectation that CD would not be associated with deaths from such causes. In general, the various sensitivity analyses (adjusted Cox model, finer caliper, and inclusion of interaction terms in the propensity score model) produced similar results to the primary analyses (PSM) (Table H in S2 Text).

### Discussion

In the PSM analyses, we observed that among live births to women in groups with low expected CD rates (Robson group 1 to 4), those who had a CD had a 25% increased mortality

rate in the first 5 years of life compared with those born vaginally. The relative increase in mortality rates was greatest (39%) during the neonatal period. Neither prelabor CD compared with unscheduled CD nor repeated CD (Robson group 5) compared with those born vaginally after a previous CD (VBAC) showed a statistically significant difference in the mortality rate up to the age of 5 years. Births with a noncephalic presentation, multiples, or preterm births (Robson groups 6 to 10) were at a reduced death rate if delivered by CD rather than if delivered vaginally.

There are no trials of CD for nonmedical reasons conducted so far, and observational [21] studies in Latin America [9] and the United States [22] observed a 2 times higher risk of neonatal death following CD without a clear medical indication compared with vaginal delivery. However, these studies did not investigate the longer-term consequences of CD up to the age of 5. Another study conducted among term, singleton births to nulliparous women in Scotland, found no association between CD and child mortality between the age of 1 and 4 years [7].

We observed excess child deaths associated with CD among births with a low expected CD rate (Robson groups 1 to 4). It remains possible that there remained a subset of babies within this low CD rate group that had complications and for whom a death subsequent to CD reflected the complication and indication for the CD. We hypothesized that such deaths would be more likely to occur in the first week of life, and our analysis of the death rate conditional on surviving the first week allowed us to explore this possibility. For all groups with a low expected rate of CD (Robson groups 1 to 4), the elevated child mortality rate remained in those who survived beyond the first 28 days of life. Beyond 1 year, the rate remained elevated for most groups but was not statistically significant. It is important to note, however, that these conditional analyses may be susceptible to selection bias. If there is a protective effect of CD, then frail live births—i.e., children at a high risk of mortality, who might previously have died in utero or the first week of life—will be more common in the CD group after the first week of life. Conversely, if the CD increases mortality risk in the first week, then frail children will be less common in the CD group beyond 1 week [23].

The increased mortality rate following CD compared with vaginal birth but not with unscheduled (emergency) CD compared with prelabor CD suggests that confounding by indication for the CD is not the explanation for the association. Rather, CD may bypass important physiological stimuli initiated by vaginal births and prevent adequate transfer of the maternal microbiome, leading to altered immunological development [5]. CD can also affect future infant health via breastfeeding because it is associated with a higher likelihood of discontinued breastfeeding before 12 weeks postpartum [24]. Another hypothesis is that different epigenetic modification of gene expression for different delivery modes (e.g., intrapartum use of synthetic oxytocin, antibiotics, or CD) affects future infant health [5].

A previous CD is a frequent indication for conducting a CD [9]. The evidence available in the literature on the effect of VBAC on child survival is controversial [6,23]. Studies have examined long-term effects of CD on offspring of women with a previous uterine scar found an increased risk of asthma, obesity, and learning disability [6], but few studies assessed mortality outcomes. Similar to a study in US women [25], we showed increased neonatal mortality among births delivered by repeated CD compared with VBAC. However, our study did not observe a difference in child survival after the neonatal period, in contrast to a study conducted in Scottish women, which observed 50% higher infant mortality [8].

We confirmed the protective association of CD on noncephalic presentation, similar to the benefits seen in observational studies and clinical trials [26,27]. Moreover, our study showed a strong protective association of CD among pregnancies with multiple and preterm births. There is limited evidence in the literature of an association between mode of delivery and

neonatal or child mortality in either multiple or preterm births [28,29]. Deaths of babies in a noncephalic presentation following a vaginal delivery can either be related to difficulty with the vaginal mode of delivery itself, such as when a foot presents first (footling) or if the fetus is large and causes a mechanical problem or experiences other problems during labor, such as birth trauma [30]. The mortality risk is particularly high in settings where medical staff are inexperienced or have inadequate skills in performing vaginal delivery, making a planned CD the safest option for such mothers and babies [31].

This study has several strengths. We used a population-based cohort with a large sample size and sufficient power to assess the rare outcome of child death. A novel feature is that our study is stratified by Robson classification as a proxy of medical indication. We conducted the sensitivity analysis to quantify residual confounding due to socioeconomic differences between women delivered vaginally and via cesarean section, and we did not find evidence suggestive of residual confounding. We also studied and compared all subgroups of deliveries rather than focusing on a small subset such as preterm or breech births.

There are, however, limitations. First, the proportion of missing data for the Robson group classification limited the analysis to 86.9% of the live births in Brazil. Second, the linkage errors may have contributed to misclassification but are unlikely to introduce bias in a consistent direction since we did not find more linkage errors by mode of delivery. However, the absolute measure of risk of death is underestimated, both due to linkage error and because some children were censored before the age of 5, as indicated by the lower under-five mortality risk observed of 11/1,000 live births, compared with what is expected (around 15/1,000 live births) [32]. Third, residual confounding is possible because data on maternal health conditions (e.g., comorbidities such as diabetes) and access to and quality of local health services were not available. Any health or service-related problem that might indicate a CD and increase the risk of child mortality should ideally have been adjusted for.

Despite these limitations, our data suggest that there could be deleterious effects from the high rates and potential overuse of CD in Brazil among women in groups with low expected rates of CD (Robson 1 to 4) and that in these groups, CD is associated with increased child mortality in the neonatal period and beyond. Pregnant women often believe CD is harmless, or even "easier" on the baby, and thus beneficial [33]; however, the findings from this study suggest that the procedure may be associated with elevated child mortality risks in certain circumstances. On the other hand, among pregnancies with strong indications for CD, the procedure remains a crucial practice for protecting children's lives; our data show that in these groups, CD is possibly being underused, particularly in low-income municipalities.

In summary, our results provide evidence that overuse and underuse of CD is associated with child survival. The study will help pregnant women and their providers make informed decisions as to whether CD is appropriate for them. There is strong demand for such information, especially in low- and middle-income settings. We recommend further research on nonmedical indicated cesarean section in low- and middle-income settings to confirm our finding that CD is associated with an increased risk of child mortality and to explore the effects of CD on child morbidity. If confirmed, interventions targeting pregnant women, health workers, and health systems should be made to reduce the rates of unindicated CD based on nonmedical grounds.

## Supporting information

**S1 Text. The RECORD statement.** RECORD, Reporting of studies Conducted using Observational Routinely-collected Data.
(DOCX)

**S2 Text. Table A:** Robson Classification and CD rate. **Table B:** SMDs before and after matching for the matching covariates, Robson group 1 to 4. **Table C:** SMDs before and after matching for the matching covariates, Robson group 5. **Table D:** SMDs before and after matching for the matching covariates, Robson groups 6 to 10. **Table E:** Logistic regression estimates of the odds of delivery by CD in Robson groups 1 to 4 or by vaginal delivery in Robson group 5 and groups 6 to 10. **Table F:** Mortality conditional on survival up to 6 days, 27 days, and under 1 year, by mode of delivery in Robson groups 1 to 4 before and after PSM, Brazil 2012 to 2018. **Table G:** Under-five mortality from external causes of death*, Brazil 2012 to 2018. **Table H:** HRs from sensitivity analyses for under-five mortality. PSM, propensity score matching; SMD, standardized mean difference.
(DOCX)

**S1 Fig.** Distribution of propensity scores by mode of delivery: **(A)** Robson groups 1 to 4, **(B)** Robson group 5, and **(C)** Robson groups 6 to 10. * Groups 2b and 4b were excluded. CS, cesarean section.
(TIF)

## Acknowledgments

We thank the data production team at CIDACS/FIOCRUZ for their work in linking these data and for providing information on data quality. We also thank the IT team for making enormous efforts to help us access the data.

## Author Contributions

**Conceptualization:** Enny S. Paixao, Christian Bottomley, Mauricio L. Barreto, Oona M. R. Campbell.

**Data curation:** Maria Yury Ichihara, Mauricio L. Barreto.

**Formal analysis:** Enny S. Paixao, Luciana L. Cardim, Flavia Jôse Oliveira Alves.

**Funding acquisition:** Laura C. Rodrigues, Liam Smeeth, Mauricio L. Barreto.

**Methodology:** Enny S. Paixao, Christian Bottomley, Julia M. Pescarini, Kerry L. M. Wong, Luciana L. Cardim, Maria da Conceicao N. Costa, Maria Gloria Teixeira, Oona M. R. Campbell.

**Resources:** Rita de Cássia Ribeiro Silva.

**Supervision:** Maria da Conceicao N. Costa.

**Visualization:** Enny S. Paixao.

**Writing – original draft:** Enny S. Paixao.

**Writing – review & editing:** Enny S. Paixao, Christian Bottomley, Julia M. Pescarini, Kerry L. M. Wong, Luciana L. Cardim, Rita de Cássia Ribeiro Silva, Elizabeth B. Brickley, Laura C. Rodrigues, Flavia Jôse Oliveira Alves, Maria do Carmo Leal, Maria da Conceicao N. Costa, Maria Gloria Teixeira, Maria Yury Ichihara, Liam Smeeth, Mauricio L. Barreto, Oona M. R. Campbell.

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
