## [Editor Report · Decision Letter 0]

22 Feb 2021

Dear Dr Paixao, 

Thank you for submitting your manuscript entitled "Caesarean deliveries and child mortality: Analysis of 17.8 million births in Brazil" for consideration by PLOS Medicine.

Your manuscript has now been evaluated by the PLOS Medicine editorial staff and I am writing to let you know that we would like to send your submission out for external peer review.

Kind regards,

Caitlin Moyer, Ph.D.

Associate Editor

PLOS Medicine

---

## [Decision Letter · Decision Letter 1]

26 Apr 2021

Dear Dr. Paixao,

Thank you very much for submitting your manuscript "Caesarean deliveries and child mortality: Analysis of 17.8 million births in Brazil" (PMEDICINE-D-21-00836R1) for consideration at PLOS Medicine. 

Your paper was evaluated by a senior editor and discussed among all the editors here. It was also sent to three independent reviewers, including a statistical reviewer. The reviews are appended at the bottom of this email and any accompanying reviewer attachments can be seen via the link below:

[LINK]

In light of these reviews, I am afraid that we will not be able to accept the manuscript for publication in the journal in its current form, but we would like to consider a revised version that addresses the reviewers' and editors' comments. Obviously we cannot make any decision about publication until we have seen the revised manuscript and your response, and we plan to seek re-review by one or more of the reviewers. 

We expect to receive your revised manuscript by May 17 2021 11:59PM. Please email us (plosmedicine@plos.org) if you have any questions or concerns.

We look forward to receiving your revised manuscript. 

Sincerely,

Caitlin Moyer, Ph.D.

Associate Editor 

PLOS Medicine

plosmedicine.org

1.Title: Please revise your title according to PLOS Medicine's style. Your title must be nondeclarative and not a question. It should begin with main concept if possible. "Effect of" should be used only if causality can be inferred, i.e., for an RCT. Please place the study design ("A randomized controlled trial," "A retrospective study," "A modelling study," etc.) in the subtitle (ie, after a colon).

2. Data availability statement: Please provide a more direct link to the specific datasets used in the study, and, if possible, a contact email address for requests for data access.

3. Abstract: Methods and Findings: Please quantify all main results presented in the abstract with both 95% CIs and p values.

4. Abstract: Methods and Findings: In the last sentence of the Abstract Methods and Findings section, please describe the main limitation(s) of the study's methodology.

5. Abstract: Conclusions: Please address the study implications without overreaching what can be concluded from the data; the phrase "In this study, we observed ..." may be useful.

6. Author summary: At this stage, we ask that you include a short, non-technical Author Summary of your research to make findings accessible to a wide audience that includes both scientists and non-scientists. The Author Summary should immediately follow the Abstract in your revised manuscript. This text is subject to editorial change and should be distinct from the scientific abstract. Please see our author guidelines for more information: https://journals.plos.org/plosmedicine/s/revising-your-manuscript#loc-author-summary

7. Throughout the text: Please place in-text citations within square brackets, before the punctuation, like this [1]. When multiple references are included, please do not include spaces within the brackets.

8. Introduction: Lines 84-85: Please revise this sentence to avoid causal implications: “... we investigated the effect of CD on child mortality…”

9. Methods: Please ensure that the study is reported according to the STROBE guideline, and include the completed STROBE checklist as Supporting Information. Please add the following statement, or similar, to the Methods: "This study is reported as per the Strengthening the Reporting of Observational Studies in Epidemiology (STROBE) guideline (S1 Checklist)."

10. Methods: Did your study have a prospective protocol or analysis plan? Please state this (either way) early in the Methods section.

11. Results: Line 173: Please define the abbreviation “PSM” at first use, and use consistently after.

12. Results: Please provide p values in addition to 95% CIs for all results presented in the text.

13. Discussion: Line 270 and 271-272: Please revise these sentences to avoid causal implications: “We confirmed the protective effect of CD on non-cephalic presentation.” and “Moreover, our study showed a strong protective effect of CD among pregnancies with multiple and preterm births.”

14. Discussion: Line 300: Please include the missing reference.

15. Discussion: Line 304-305: Please revise this sentence to avoid implying causality: “In summary, our results provide evidence that either overuse or underuse of CD can affect child survival adversely.”

16. References: Please use the "Vancouver" style for reference formatting, and see our website for other reference guidelines https://journals.plos.org/plosmedicine/s/submission-guidelines#loc-references

17. Table 1: Please define abbreviations “HDI” and “CD” in the legend. Please confirm if CS should be CD (as used elsewhere in the manuscript). Please include in the legend that values are number and percent.

18. Table 2: Please include p values for both the unmatched and propensity score matched results. In the left hand column, please indicate these are Robson groups.

19. Table 3: Please define the abbreviation “CD” in the legend, and please indicate the left hand column represents the Robson group. Please also provide the p values in addition to the 95% CIs for the adjusted and unadjusted analyses.

20. Figure 1: We suggest a more descriptive title for this figure would be helpful.

21. Figure 2: Please include a legend describing the six panels, A-F, and we suggest a more descriptive title.

22. Table S1: Please define the abbreviation “CS”

23. Table S2, S3, and S4: Please define the abbreviation “HDI” and “CS” in the legend.

24. Table S5: Please provide p values in addition to 95% CIs. Please define “HDI” and “CS” in the legend.

25. Table S6: Please include p values in addition to 95% CIs. Please indicate in the legend that “before” and “after” refer to propensity score matching.

26. Table S7: Please include p values in addition to 95% CIs.

27. Table S8: Please revise the title to avoid causal language. Please include p values in addition to 95% CIs. Please define the abbreviation “PSM” in the legend.

Comments from the reviewers:

Reviewer #1: I confine my remarks to statistical aspects of this paper. These were well done, and I have only some minor points to address before I can recommend publication.

General

It might be useful to give deaths per capita (or per 100,000 babies or something) in addition to per person year. Person-year is appropriate statistically, but not very intuitive. Doctors and patients probably want to know "What is the risk that the baby dies?" One way to address this is to do logistic regression in addition to Cox. However, a rough estimate of this can (if I am thinking straight) be gotten by dividing the person-year figure by about 5. 

Specific

p. 2 line 41-2 Insert "significant" There was some evidence

Table 1 I would give the percentage row-wise rather than column-wise. I am more interested in, e.g. what proportion of group 1 were vaginal vs. CD rather than what proportion of vaginal births were group 1

Table 2: Not surprisingly, nearly all the deaths were in the risky groups. I think more could be said about this. For a doctor and patient trying to decide whether to have vaginal delivery or CD, the key factor is not the proportion of added risk in odds ratio or hazard ratio terms, but the actual risk of the baby dying. Going from a risk of 1 in 2000 to a risk of 1 in 1800 is not the same as going from 1 in 20 to 1 in 18. Some of this can be seen in fig 2, 

Non-statistical point of interest: What about women who had CD for a reason related to their own health? For instance, my ex-wife had CD because she has a shunt in her belly and the ob-gyn thought that labor might induce gastrointestinal fluid to go into her brain. I have no idea how common such things are, but if they are at least somewhat common, they might be worth a comment - they don't seem to be captured in Robson.

Peter Flom

Reviewer #2: This is a really very interesting manuscript and in my opinion, it should be published. 

The introduction is well written, the methods are presented clearly. The statistical analyses are correct. The results are presented very clearly and the results are critically discussed. 

There are only few points that should be clarified.

1) there is no hypothesis presented, which is tested in tthis study. Please add a clear hypothesis. 

2) The sample is not described sufficiently. Please indicate inclusion and exclusion criterions

3) Please add a table containing the sample characteristics. maternal age, sociodemographioc characteristics, reproductive history, indepnendt of Robson criterions. Just for the whole sample. 

Reviewer #3: Excellent paper addressing an important public and reproductive health topic, that is the Caesarean Section epidemic, in particular in a country with one of the highest prevalences worldwide.

The strength of the paper is based on 1) huge volumes ( >17 million deliveries in one country over 6 years); 2) a national vital registration system providing a unique opportunity to analyse data; and 3) a very well designed methodology based on the Robson classification and using prospensity score matching to address some of the key questions.

The objectives are clear, the questions addressed very relevant, the methodology well described, analyses well done, results clear and the discussion well written.

Limitations are also addressed.

In summary; this paper deserves publication accompanied by a clear message in the format of an editorial or a comment.

Few suggestions:

- would be interesting to examine maternal mortality/near miss morbidity

- could be followed by a paper on how many lives saved modelling these findings to other national or regional data

some small typos

- line 177 scoire

- line 194; should be if instead of it

-line 301 at risk.Or

[LINK]

---

## [Decision Letter · Decision Letter 2]

2 Aug 2021

Dear Dr. Paixao,

Thank you very much for re-submitting your manuscript "Caesarean deliveries and subsequent child mortality: a national record-linkage longitudinal study of 17.8 million births in Brazil" (PMEDICINE-D-21-00836R2) for review by PLOS Medicine.

I have discussed the paper with my colleagues and the academic editor and it was also seen again by two reviewers. I am pleased to say that provided the remaining editorial and production issues are dealt with we are planning to accept the paper for publication in the journal.

[LINK]

We look forward to receiving the revised manuscript by Aug 09 2021 11:59PM.   

Sincerely,

Caitlin Moyer, Ph.D.

Associate Editor 

PLOS Medicine

plosmedicine.org

Requests from Editors:

1. Title: Please capitalize the first word of the subtitle, and we suggest revising to: “Associations between caesarean delivery and subsequent child mortality: A national record-linkage longitudinal study of 17.8 million births in Brazil” or similar.

2. Abstract: Background: Please clearly state the study hypothesis as the final sentence.

3. Abstract: Methods and findings: Please check the p value reported for this finding: “For the live births to women with a CD in a prior pregnancy (Robson group 5), the relative rates for child mortality were similar for those born by CD compared with vaginal deliveries (HR=1.05, 95% CI:1.00-1.10; p=0.024).”

4. Abstract: Conclusion: We suggest replacing “the benefits of clinically-indicated CD were confirmed.” with “...we observed a reduction in child mortality.” As Robson group was used as a proxy for CD indications, we suggest tempering the language here: “These findings may suggest that clinically-indicated CD is associated with child outcomes.” or similar.

5. Author summary: Why was this study done?: Please revise the third point to “Unindicated CDs add to costs, potentially introduce harms, and the effects of unindicated CD on infant and child health outcomes remain unclear, especially in low and middle income countries.”

6. Author summary: Please use the abbreviation "CD" throughout after it is first introduced.

7. Author summary: What did the researchers do and find?: Please combine the first two bullet points. Please clarify the third bullet point to make it clear if this is among women with a previous CD, or among all live births.

8. Author summary: What do these findings mean? Please revise the first bullet point to remove causal implications: “Our study suggests that, in Brazil, CD can be associated with increased risk of child mortality unless there is a clear indication for the procedure.”

9. Methods: Page 7-8: Please identify the confounders included in the propensity score model and adjusted for in the Cox proportional hazards model, and also include the information on how each was reported/assessed and categorized for the analysis.

10. Methods: Please clarify how statistical significance of differences between comparison groups was determined.

11. Results: “The mortality risks between vaginal and caesarean deliveries were not so

markedly different in groups 1-5 (Table 2).” Please make it clear in the text if these differences were statistically significant or not.

12. Results: “In a breakdown by time-period, unscheduled CD was associated with increased rate during the neonatal period (HR=1.18, 95% CI: 1.05-1.32; p=0.003) in group 2 and in the first year of life (HR=1.15, 95% CI: 1.03-1.29; p=0.010) in group 4 (Table 3).” Please clarify if “associated with an increased mortality rate during the neonatal period…” is meant.

13. Discussion: First sentence: We suggest beginning with “In a propensity score-matched analysis, we observed that among live births to women with low expected CD rates (Robson group 1-4), who had a CD…”

15. Discussion: “We conducted the sensitivity analysis to quantify residual confounding due to socioeconomic differences between women delivered vaginally and via caesarean section and confirmed there was little.” Please revise this sentence, as the findings of the sensitivity analysis do not confirm that there was little residual confounding. Rather, the sensitivity analysis did not find evidence suggestive of residual confounding, but that does not confirm that residual confounding may not be present.

16. Conclusion: “We recommend further research in low- and middle-income settings to confirm our finding that CD increases the risk of child mortality…” Please revise this sentence to avoid causal implications.

17. Competing interests, Financial Disclosures and Data Availability: Please remove these sections from the main text of the manuscript. This information will be taken directly from the relevant sections of the manuscript submission metadata.

18. References: Please double check each reference for the correct formatting. Please use the correct abbreviations for journal titles (e.g. “Lancet” vs “The Lancet”). Please use the "Vancouver" style for reference formatting, and see our website for other reference guidelines https://journals.plos.org/plosmedicine/s/submission-guidelines#loc-references

19. Figure 2 and Figure S1: Please provide descriptive legends for each of these figures.

20. Supporting information Table S3: Please include this table in the main text of the paper, rather than as a supporting information table.

21. Table S10: Please clarify in the legend whether the “primary analysis” represents the Propensity Score Matched analysis.

22. We suggest including the RECORD checklist as a separate file.

Comments from Reviewers:

Reviewer #1: The authors have addressed my concerns and I now recommend publication.

Peter Flom

Reviewer #2: in my opinion, the authors have addressed all suggestions of the reviewers. 

Therefore, I recommend to accept this manucscript in its present form.

[LINK]

---

## [Editor Report · Decision Letter 3]

2 Sep 2021

Dear Dr Paixao, 

On behalf of my colleagues and the Academic Editor, James Tumwine, I am pleased to inform you that we have agreed to publish your manuscript "Associations Between Caesarean Delivery and Child Mortality: A National Record-Linkage Longitudinal Study of 17.8 Million Births in Brazil" (PMEDICINE-D-21-00836R3) in PLOS Medicine.

In addition, please address the following editorial issues:

1. Data availability statement: Please add a more direct link, or a contact email address (such as cidacs.comunicacao@fiocruz.br, if accurate) to help interested parties to request access to the study data on the CIDACS website.

2. References: The reference list is not formatted correctly. Please consistently format the references according to the "Vancouver" style for reference formatting (guidance and examples can be found at: https://journals.plos.org/plosmedicine/s/submission-guidelines#loc-references). Specifically, please list up to the first six authors (as last name followed by initials, followed by et al. if there are more than six authors. Please use the appropriate NLM abbreviations for journal titles (for example “Lancet” instead of “The Lancet”) and please do not use italics or place the article titles in quotations.

PRESS

Sincerely, 

Caitlin Moyer, Ph.D. 

Associate Editor 

PLOS Medicine